# Weed community changes in saffron +chickpea intercropping under different irrigation management

Fatemeh Mohammadkhani[1], Majid Pouryousef[1]*, Ali Reza Yousefi[1], Jose L. Gonzalez-Andujar[2]

1 Department of Plant Production & Genetics, University of Zanjan, Zanjan, Iran, 2 Instituto de Agricultura Sostenible (CSIC), Córdoba, Spain

* pouryousef@znu.ac.ir

## Abstract

Saffron (*Crocus sativus* L.) is among the world's most expensive crops; nevertheless, it struggles to compete with weeds. Non-chemical farming practices, such as intercropping and reduced irrigation, can help to decrease weed problems. Therefore, this study aimed to evaluate the changes in the weed density, biomass and weed diversity under saffron-chickpea intercropping system with two irrigation regimes. The study's treatments included two irrigation regimes, namely one-time irrigation and conventional irrigation (carried out four times from October through May), and six planting ratios of saffron and chickpea, namely saffron sole-crop (C1), chickpea sole-crop (C2) in eight rows, 1:1 (C3), 2:2 (C4), 2:1 (C5), and 3:1 (C6)] as main and sub-plots, respectively. The result showed that the conventional irrigation regimes increased weed diversity, however, it didn't affect the Pielou index. Intercropping ratios decreased weed diversity compared to saffron and chickpea mono-cropping systems. The interaction effect of treatments was significant for weed density and weed biomass. In most intercropping ratios, weed density and weed biomass decreased under one-time irrigation regimes. The lowest values for weed density and biomass were observed with an average of 15.5 plants/m$^2$ and 37.51 g/m$^2$, respectively, under the one-time irrigation regime with C4 intercropping systems. This intercropping system did not show a significant difference with C3. Overall, the results indicate that a one-time irrigation regime and intercropping with chickpea, specifically with a 1:1 saffron-chickpea ratio (C3) and a 2:2 saffron-chickpea ratio (C4), could be effective strategies for weed management in saffron in semi-arid cropping systems.

## Introduction

Saffron (*Crocus sativus* L.) is one of the most expensive spice, medicinal and industrial crops worldwide [1]. The plant is used as a spice to flavor food, and fragrance in perfumes and cosmetics worldwide. It has been used in traditional medicine as an anti-depressant, anti-carcinogen, and stimulant, along with a wide range of other applications [2, 3]. The main saffron-

**Data Availability Statement:** All relevant data are within the manuscript.

**Funding:** The author(s) received no specific funding for this work.

**Competing interests:** The authors have declared that no competing interests exist.

producing countries are Iran, India, Greece, Afghanistan, Morocco, Spain, Italy, China, and Azerbaijan [4]. Iran is the leading country in saffron production [5], with 120,000 ha, and 3.2 kg ha$^{-1}$ as cultivated area, and average yield, respectively [6]. Iran produces almost 90% of global, saffron [7].

Saffron is cultivated in different climatic conditions, for example, in southern Europe, North Africa, the Middle East, and Central Asia, which have low rainfall, warm summers, and cold winters [8]. It has a lower water requirement than many other crops due to its morpho-physiological properties [9] and drought tolerance and having a dormant stage (from May to August) [10]. Nevertheless, irrigation increases saffron yield; and is therefore irrigated four or five times from October through May [11]. de Juan, Córcoles [12] reported that irrigation in March and April, as well as in August and September, increases saffron flower and stigma yield. However, Koocheki and Seyyedi [13] indicated that in western Macedonia irrigation increased leaf development, but reduced the number of flowers and quality of saffron.

After flowering in October, saffron grows actively through April and then the plant senesces, leaves wither (April–May), and corms become dormant approximately until July. Therefore, the fields are without any crop canopy during this time, and weeds can occupy the space and suppress the saffron plant. Therefore, weeds are the main problem in saffron production, since this plant is a poor competitor with weeds and grows slowly, so weed growth at any stage of crop growth can hurt the quality and quantity of the product. Weeds are controlled by mechanical, hand weeding, or chemical methods. Although mechanical and hand weeding are effective and environmentally friendly, they are time-consuming and costly [14, 15]. Hosseini-Evari et al. [16] revealed that a combination of herbicide (Metribuzin) with the mulch method can be useful for weed management in saffron. Chemical weed control is too expensive and has negative side effects on the environment and human health [17].

Irrigation management is one of the strategies of weed control for the development of a weed management system, especially in arid and semiarid regions. Therefore, a low irrigation schedule during the weed life cycle could reduce weed germination and presence by less water availability for weed growth. Metwally et al. [18] stated that in peanuts, total weed dry weight reduced up to 25.66–25.71% because of decreased irrigation by 60%. Verma et al. [19] reported that weed density and biomass were reduced at the lower rate of irrigation because of the less availability of water.

Another strategy for preventing weed dominance is using intercropping systems. This cultivation system, the simultaneous cultivation of one or multiple crop species in the same field area, is a good practice to enhance productivity through efficient use of nutrients, light, and water while suppressing weeds [20–23]. Moghaddam et al. [24] showed that saffron-white bean intercropping, concurrent with 10 mM glycine betaine application, can play a pivotal role in regulating soil temperature and increasing land equivalent ratio. Some evidence showed that intercropping can decrease weeds' germination, emergence, and flowering [25–27]. Shabahang et al. [28] reported that *Vicia faba* with saffron intercropping decreased weed density by 93% as compared to weed-infested plots. Shah et al. [29] showed that bean-rice intercropping reduced weed presence by 65%. Saudy [30] reported that maize–cowpea intercropping suppressed weeds over all N levels, reducing weed biomass by 49.5% compared to sole-maize. Bilalis et al. [31] pointed out that intercropping maize with legumes significantly decreased weed density compared with mono-cropping corn by reducing light access for weeds.

Research on the effects of intercropping on weed community structure (or diversity) is limited. Intercropping can change weed community structure [32–34], mainly due to the increasing emergence of the spring species [34]. Berdjour et al., [35] reported that intercropping corn-soybean decreased weed species diversity by 40% compared to the sole-crop. Sharma & Banik, [36] showed a weed diversity reduction by the intercropping system.

Legumes represent about 27% and 15% primary agricultural production and proteins worldwide, respectively, [37]. Legumes are considered a key species in intercropping systems due to their ability to fix nitrogen. In this study, chickpea (*Cicer arietinum* L.), an small annual legume from the Fabaceae family, was used as the companion crop. Chickpeais the third most cultivated legume in the world after dry bean (*Phaseolus vulgaris*) and peas (*Pisum sativum*) [38]. This plant is primarily cultivated in arid and semi-arid regions, and it plays an important role in intercropping systems due to its ability to fix nitrogen [39], reduce the need for chemical inputs, and improve soil fertility [40, 41]. Abbasi and Sepaskhah [42] reported that farmers can use the inter-row spaces of the saffron crop to grow winter wheat simultaneously, resulting in higher revenue and land productivity, as well as clean and sustainable production.

Saffron planted in arid and semiarid regions, and is a poor weed competitor, resource limitation like water, and the application of intercropping systems could be effectively in weed control saffron fields. To the best of our knowledge, there is no study on the effect of the saffron–chickpea intercropping system on weed populations. Therefore, the present research aimed to study the effects of saffron+chickpea intercropping on weed control and community structure (or diversity) under two irrigation regimes.

## Material and methods

### Site description

The present study was performed for two growing seasons from 2018–19 to 2019–20 at the Research Farm of the Faculty of Agriculture, University of Zanjan (latitude 36° 41′ N, longitude 48° 27′ E; 1620 m above sea level), Zanjan, Iran. The climate of this region is classified as semi-arid cold, with an average annual temperature of 11°C and average annual precipitation of 293 mm (Fig 1). The soil characteristics of the experimental site were as follows: sandy-loam texture, 7.54 pH, and 1.1% organic matter.

### Experimental design and treatments

The field experiments were arranged in a split-plot pattern with three replications. Each plot was 2m × 2m in size.

In mono-cropping, saffron was planted with a distance of 25 cm between the rows and 7 cm between the plants on the row and at a depth of 15–20 cm, which was a total of 80 plant/m$^2$. Chickpeas were also cultivated with a density of 50 plants/m$^2$ with a depth of 3 to 4 cm.

The first factor included two drip irrigation regimes i.e. one-time irrigation and conventional irrigation as main plots which the size of each main plot was 14.5×2 m and size of subplots was 2m × 2m. Before planting, a disk and leveler were used to prepare the land for cultivation, and sowing data was in the middle of the summer of 2017. After preparing the land and preparing the seedbed, 30 tons of rotted cattle manure was applied to the land and mixed thoroughly with the soil, and no other fertilizers were applied to the soil over the experiment. The nutrient management practices were kept consistent across all treatments.

A one-time irrigation treatment was applied following saffron planting in October, and conventional irrigation was applied four times from October through May based on rainfall conditions (see Table 1). The second factor consisted of six planting ratios of saffron: chickpea, (C1) saffron mono-crop, (C2) chickpea mono-crop, (C3) 1saffron+1chickpea, (C4) 2saffron +2chickpea, (C5) 2saffron+1chickpea, and (C6) 3saffron+1chickpea; details are presented in Fig 2. Saffron corms and chickpea seeds were planted manually in 25-cm row spacing (80 and 50 plant/m$^{-2}$, respectively), in early October and March 2018, respectively (Fig 2). No herbicides or pesticides were applied throughout the experiment. Also, chemical and physical soil characteristics was indicated in Table 2.

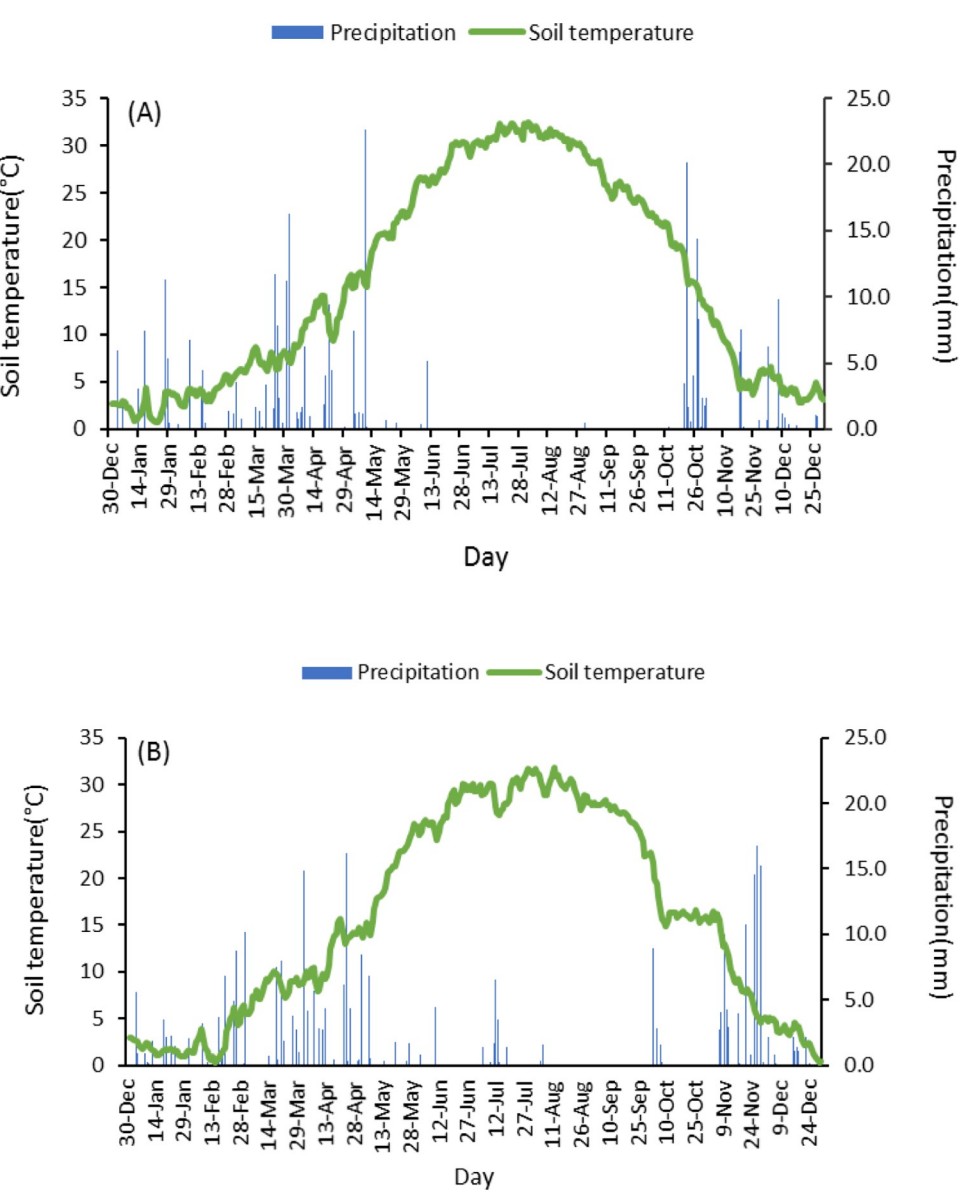

**Fig 1.** Average daily soil temperature (˚C) at 9 cm and precipitation in 2019 (A) and 2020 (B). Meteorological data from Zanjan University Meteorological Station (from 2018–19 through 2019–20), available in (http://www.znu.ac.ir/weather).

## Measurements and sampling

Aboveground weeds were hand collected from 1 m$^2$ area in each subplot at 75 days after sowing (DAS) of the chickpea in 2019 and 2020. Weed species were identified and counted based on the photosynthetic system. Weed biomass was oven-dried at 63˚C for 72h and dry weights

**Table 1. Water quantity and the time of application.**

| Irrigation regimes | First irrigation | Second irrigation | Third irrigation | Forth irrigation | Quantity of water applied (M$^3$ ha$^{-1}$) |
|---|---|---|---|---|---|
| One-time irrigation | Early-October | - | - | - | 700 |
| Conventional irrigation | Early-October | Early-March | Late-April | Mid-May | 2800 |

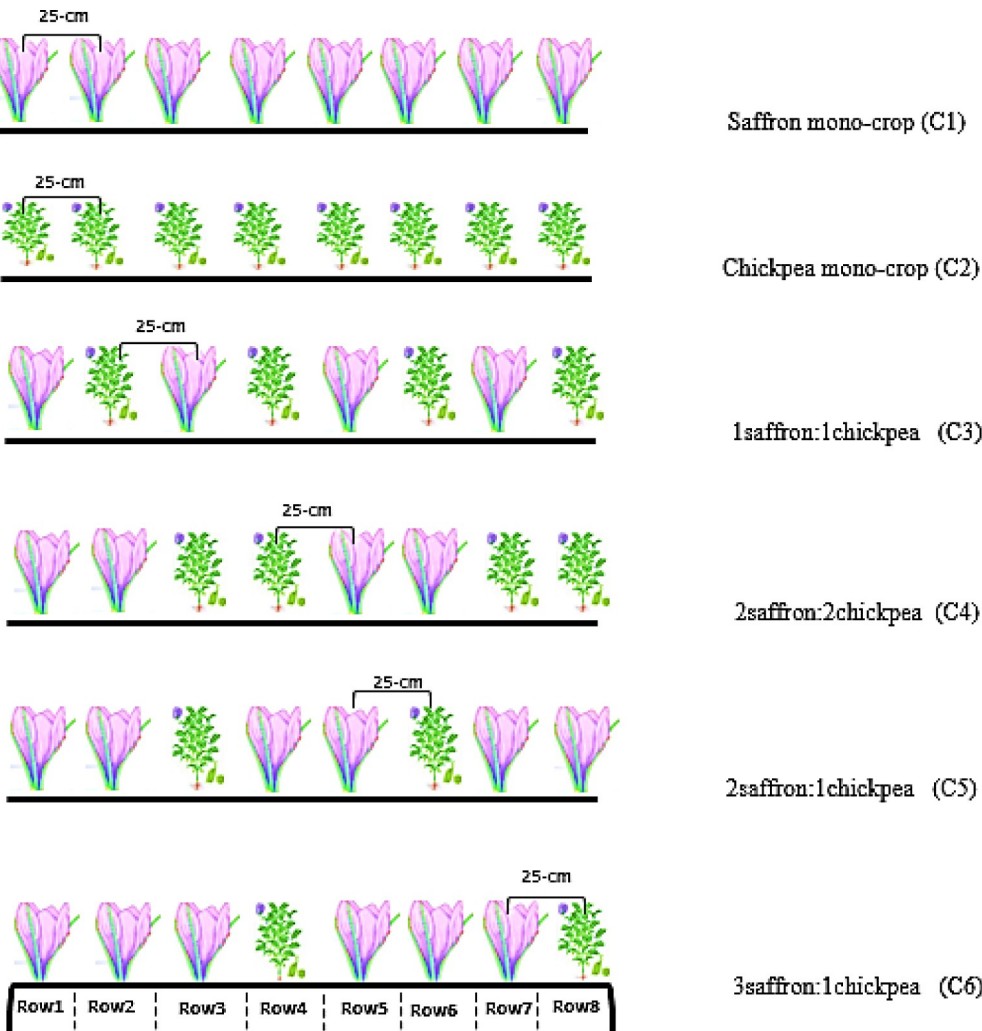

**Fig 2. Schematic diagram of saffron and chickpea intercropping based on row replacement series.** (C1) saffron mono-crop, (C2) chickpea mono-crop, (C3) 1saffron:1chickpea, (C4) 2saffron:2chickpea, (C5) 2saffron:1chickpea, and (C6) 3saffron:1chickpea, respectively. Saffron corms and chickpea seeds were planted with 25-cm row spacing.

(g m$^{-2}$) were determined. For sampling, in each plot were selected area after ignoring 0.5 m from each sides of plots.

To assess weed diversity, Species richness (S), the Exponential Shannon Index, and Pielou's Evenness Index were calculated for each plot and year separately. Species richness (S) is the weed species' total number. the Exponential Shannon Index (eH; [43], was estimated as:

$$Eh = exp\left[-\sum\nolimits_{i=1} pi\, Ln\,(pi)\right]$$

Where Eh is the exponential Shannon Index, pi is the relative abundance of the i species in the community, and Ln is the natural logarithm.

**Table 2. Physical and chemical properties of farm soil.**

| Organic Carbon (%) | Total nitrogen (%) | Phosphorus (ppm) | Potassium (ppm) | pH | EC (ds/m) | Soil texture |
|---|---|---|---|---|---|---|
| 0.85 | 0.092 | 13.4 | 283 | 7.94 | - | Clay loam |

Pielou's Evenness Index (J′; [44] was calculated as;

$$J' = \left[ \frac{H'}{H'_{Max}} \right]$$

Where J′ is Pielou's Evenness Index, H is the number derived from the Shannon diversity index and H′ max is the maximum possible value of H' (if every species was equally present).

## Statistical analysis

Statistical analysis was performed using PROC GLMMIX in SAS (version 9.4). The data were first checked for normal distribution and variance homogeneity using the Shapiro–Wilk test (PROC UNIVARIATE) and the Bartlett test (PROC GLM), respectively. Weed density (plant $m^{-2}$) and biomass (g $m^{-2}$) were square rooted and log-transformed, respectively. Mean comparisons were done using the least significant difference (LSD) test at a 95% confidence level (P ≤ 0.05). Because there was a significant effect of year and there were several treatment interactions data were analyzed and presented separately by the year.

## Results and discussion

Based on the results of combine analysis of the effect of irrigation regimes (I), and intercropping system (C) on weed control and weed diversity properties in Table 3, the main effect of year were significant at 5 and 1% probability for Exponential (Shannon Index) and Species richness, respectively. Also, the main effect of irrigation were significant at 1, 1, 5, and 5% probability for Weed density, Weed biomass, Exponential (Shannon Index) and Species richness respectively. The mine effect of intercropping was the same as irrigation, the only different between that is Pielou Index was significant at 5% probability in this effect. About interaction effect of IR×CR weed density and weed biomass were significant at 5% probability level (Table 3).

### Weed flora

A total of nine weed species were identified during both years of the experiment, and only two species had a perennial life cycle. Two weed species belonging to Poaceae families, had a $C_4$

**Table 3. Combine analysis of the effect of irrigation regimes (I), and intercropping system (C) on weed control and weed diversity properties.**

| S.O.V | Df | MS | | | | |
|---|---|---|---|---|---|---|
| | | Weed density | Weed biomass | Exponential (Shannon Index) | Pielou Index | Species richness |
| Year(Y) | 1 | 0.0043[ns] | 0.0003[ns] | 0.0990* | 0.0051[ns] | 37.5555** |
| Repeat(Y) | 4 | 0.1782 | 0.0020 | 0.0438 | 0.0077 | 0.9722 |
| Irrigation(IR) | 1 | 1.6200** | 0.0690** | 0.3667* | 0.0042[ns] | 22.2222* |
| Y*IR | 1 | 0.0150[ns] | 0.0000[ns] | 0.0049[ns] | 0.0007[ns] | 2.0000[ns] |
| Repeat*IR(Y) | 4 | 0.0236 | 0.0010 | 0.0100 | 0.0033 | 0.1944 |
| Intercropping(CR) | 5 | 2.3118** | 0.0543** | 0.1279* | 0.0051* | 7.6222* |
| IR*CR | 5 | 0.5203* | 0.0267* | 0.0114[ns] | 0.0009[ns] | 1.3555[ns] |
| Y*CR | 5 | 0.0346[ns] | 0.0006[ns] | 0.0018[ns] | 0.0014[ns] | 0.4888[ns] |
| Y*IR*CR | 5 | 0.1134[ns] | 0.0006[ns] | 0.0170[ns] | 0.0022[ns] | 0.4000[ns] |
| Error | 40 | 0.0014 | 0.0813 | 0.0182 | 0.0018 | 0.4000 |

Ns, not significant

*, **: significant at P≤0.05 and 0.01, respectively.

**Table 4. Weed flora composition of the experimental site.**

| Weed species | Common name | Family | Life Cycle | Photosynthetic pathway |
|---|---|---|---|---|
| *Convolvulus arvensis* L. | Bindweed, Field | Convolvulaceae | Perennial | C3 |
| *Chenopodium album* L. | Lambs quarters, Common | Chenopodiaceae | Annual | C3 |
| *Polygonum aviculare* L | Knotweed, Prostrate | Polygonaceae | Annual | C3 |
| *Descurainia sophia* L. | Flixweed, Herb-sophia | Brassicaceae | Annual | C3 |
| *Cuscuta compestris* Yunck | Field dodder, Golden dodder | Convolvulaceae | Annual | C3 |
| *Rumex crispus* L. | Dock, CurlY | Polygonaceae | Annual | C3 |
| *Alyssum hirsutum* M. Bieb | Garlic | Brassicaceae | Annual | C3 |
| *Cynodon dactylon* L. | Bermuda grass | Poaceae | Perennial | C4 |
| *Echinochloa crus-galli* L. | Cockspur, Barnyard millet | Poaceae | Annual | C4 |

photosynthetic pathway, whereas the other remaining species had a $C_3$ photosynthetic pathway (Table 4).

## Weed density and weed biomass response to irrigation and intercropping

The application of one-time irrigation significantly reduced total weed density compared to conventional irrigation. A similar pattern was observed in the weed biomass, where minimum weed biomass with an average 37.51 g m$^{-2}$ was obtained by the one-time irrigation regime and C4 intercropping system. Therefore, the one-time irrigation regime provided greater weed control.

According to the results, intercropping reduced both weed density and weed biomass compared to the saffron and chickpea mono-crops (C1 and C2). The treatment C4 (2:2 ratio) and C3 (1:1 ratio), were effective in reducing weed density and weed biomass. The interaction of irrigation regimes and intercropping systems showed that one-time irrigation with strategy C4 provides more control of weeds in terms of weed density and weed biomass and obtained with an average 37.51 and 15.5 respectively (Fig 3A and 3B). In summary, the results indicated that the one-time irrigation regime and the intercropping system (strategy C4) can effectively control weeds.

Crop management practices shape the weed community, influencing the productive performance of the crops. Among these practices, irrigation play a crucial role in shaping the

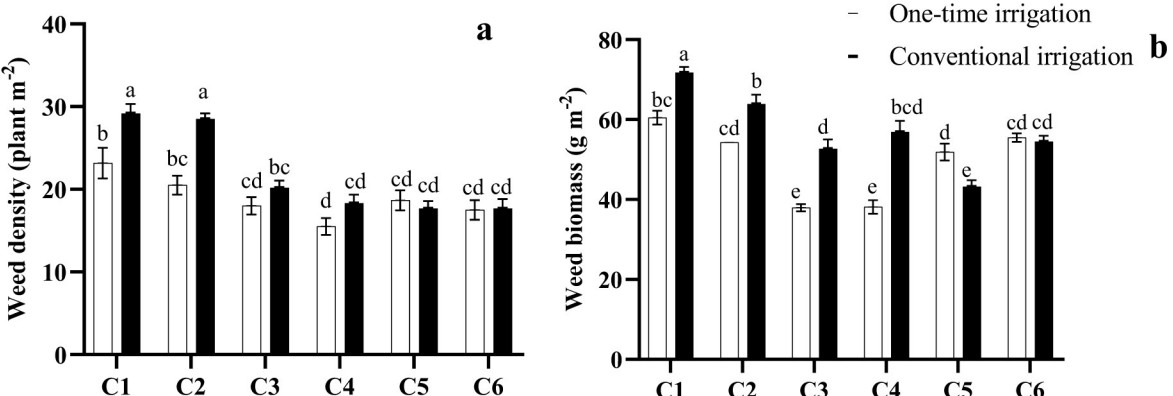

**Fig 3.** Interaction effect of irrigation regimes and intercropping system on weed density (a) and weed biomass (b) by two years. Different letters indicate a significant difference (LSD's significant difference test, P≤0.05). Planting ratios of saffron:chickpea, (C1) saffron mono-crop, (C2) chickpea mono-crop, (C3) 1saffron:1chickpea, (C4) 2saffron:2chickpea, (C5) 2saffron:1chickpea, and (C6) 3saffron:1chickpea, respectively.

composition of the weed flora particularly in semi-arid regions. Our research findings are consistent with those of other studies, which have also demonstrated that a one-time irrigation treatment was more effective in reducing weed density and biomass [45]. This superiority of the one-time irrigation regime as a weed control tool can be attributed to a decrease in weed emergences due to water availability during the weed life cycle.

Water availability is an important topic in weed emergence and growth in agriculture and weed management. The changes in water availability during the germination period of weed affect the seedling emergence pattern, reducing weed density and biomass [46]. Verma, Singh [19, 47] reported that low irrigation decreased *Chenopodium album* biomass due to an insufficient supply of water. This suggests that water stress during the germination period can limit the growth and survival of weed seedlings, ultimately reducing weed density and biomass. ud Din, Ramzan [48] reported higher density and biomass values for *Rumex Crispus*, *Convolvulus Arvensis*, and *Ammi Visnaga* L. in plots with 100% irrigation Overall, these findings highlight the importance of understanding the role of water availability in weed management strategies. By optimizing irrigation practices, farmers and weed managers can potentially reduce the emergence and growth of weeds, ultimately improving crop yields and reducing the need for herbicide applications.

The results of our study indicate that intercropping treatments in both irrigation regimes reduced weed density and biomass compared to mono-cropping. This finding is consistent with previous research which have also reported reductions in weed density and biomass in intercropping systems [49, 50].

Weed density and biomass reductions can be attributed to several factors, including competition for resources such as water, nutrients, and light, which are limited in agro-ecosystems. Specifically, the shading effect of the intercropping system can play a significant role in reducing weed growth, as higher solar radiation is intercepted by the canopy, therefore limiting the amount of light availability for weed growth [31]. Katsaruware [51] reported that intercropping maize with cowpea decreased weed biomass as compared to sole-crop, and limited the accessibility of resources for weeds such as photosynthetically active radiation (PAR) reaching the ground. Similarly, Eskandari [52] found that intercropping of wheat with fava-bean was more effective at controlling weeds compared to the wheat mono-crop. Studies have also shown that intercropping can be effective in reducing weed density and biomass in a variety of crop combinations. Banik, Midya [39] reported that wheat–chickpea intercropping could decrease weed density and biomass. Szumigalski and Van Acker [53] showed that intercropping wheat with canola or pea was more effective at suppressing weeds compared to their mono-crop. Overall, these findings highlight the potential benefits of intercropping as a sustainable weed management strategy. By reducing weed density and biomass, intercropping can help to improve crop yields, reduce the need for synthetic herbicides, and promote sustainable agricultural practices that prioritize environmental health and biodiversity.

### Weed diversity response to irrigation and intercropping

According to the results, the conventional irrigation increased the weed diversity in the plots compared to the one-time irrigation (Fig 4). There were no significant differences between the two irrigation regimes in terms of the Pielou index (equity, Fig 4).

The intercropping treatments reduced the weed diversity compared to the mono-crop treatments (C1 and C2 strategies). Species richness was reduced by C3 and C4 systems (Fig 5). There were no significant differences between the treatments in terms of the Pielou index in the 2020 crop season. The C5 and C6 systems showed a significant reduction in the Pielou index compared to the monoculture systems during the 2019 crop season. The C4 treatment

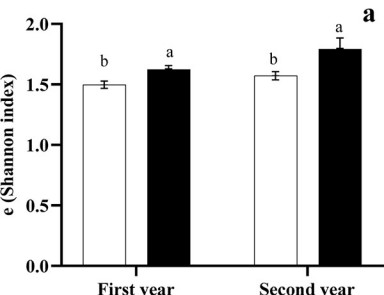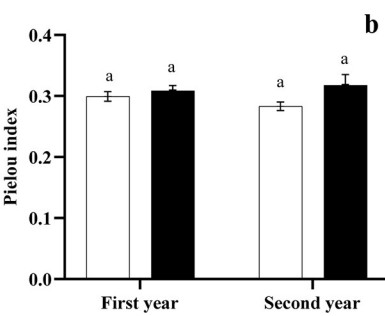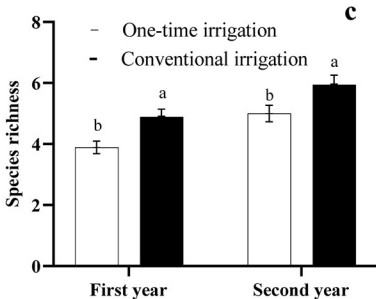

**Fig 4.** Bar plots of mean values of both, the (Shannon index) (a), Pielou index (b), and species richness (c) for the irrigation regimes. Different letters indicate a significant difference (LSD's significant difference test, P≤0.05).

significantly reduced the exponential Shannon index in both years, compared to the saffron and chickpea mono-crop treatments (Fig 5). A significant difference was observed between years in terms of the e(Shannon index) and species richness, where the maximum stated values were recorded during the second year (Fig 6).

In agricultural systems with less water availability, the dominance of weeds more adapted to water stress conditions may intensify competition, reducing agrosystems' diversity [54]. Water-limited environments often favor crop species that are better adapted to these conditions. These crops may be better able to compete with weeds for the limited water resources, reducing the growth and reproduction of weed species. Higher levels of weed species richness and diversity (exponential Shannon index) have been found under the conventional irrigation treatment compared to the one-time irrigation. It seems conventional irrigation may create a more favorable environment for weed growth and diversity through the application of regular and consistent water, allowing a greater number of weed species to establish and thrive. Additionally, conventional irrigation may increase the availability of nutrients in the soil, which can also favor weed growth and diversity. Some studies have stated that the once water supply restricts the establishment of species that are not tolerant to drought stress conditions [55–57]. Nevertheless, the evenness index was similar in both irrigation systems. As the degree of evenness in plant communities is mediated by competitive interactions [58], our results suggested that no large differences have arisen in weed composition. It has been shown by different researchers that intercropping alters the structure of weed communities [32, 34]. A significant difference between years in terms of the e(Shannon index) and species richness might be due to relatively higher precipitation and less soil temperature during this year.

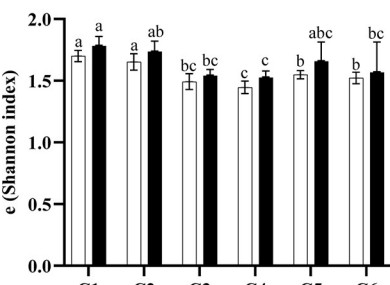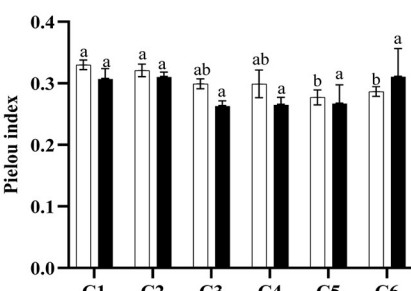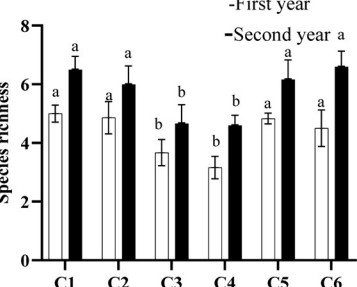

**Fig 5.** Bar plots of mean values of both, e(Shannon index) (a), Pielou index (b) and species richness (c) for the intercropping systems in 2019 and 2020 years. Different letters indicate a significant difference (LSD's significant difference test, P≤0.05). Planting ratios of saffron:chickpea (C1) saffron mono-crop, (C2) chickpea mono-crop, (C3) 1saffron:1chickpea, (C4) 2saffron:2chickpea, (C5) 2saffron:1chickpea, and (C6) 3saffron:1chickpea, respectively.

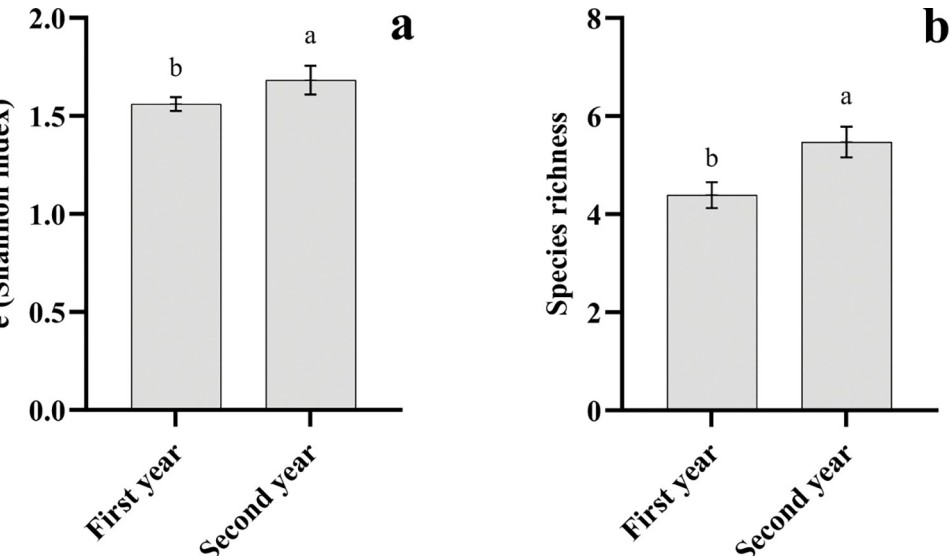

**Fig 6.** Bar plots of mean values of both the (Shannon index) (a) and species richness (b) for the year effect. Different letters indicate a significant difference (LSD's significant difference test, P≤0.05).

The species richness and diversity were found to be lower in the intercropping treatments compared to the mono-cropping treatments. One reason for this phenomenon could be the competitive advantage of the crop plants over weeds in the intercropping system due to higher solar radiation interception by the crop plants and reduced light availability for weeds.

Widaryanto [59] found that a decreased light intensity received by weeds due to intercropping reduced the diversity of species in the weed communities. Baumann et al. [60] reported that intercropping of leeks increased weed suppression due to increased light absorption by vegetation, which ultimately reduced the number of weeds.

Differences in the species richness of intercropping systems over experiments can be demonstrated by capturing light available from the intercropping system canopy. The intercropping system canopy limits the amount of light that reaches the soil surface, reducing the availability of light for weed growth. This can result in weed suppression and a reduction in weed diversity. Similar results have been reported in previous studies, which have shown intercropping can effectively suppress weed growth due to the shading effect of the crop canopy [31, 33, 61].

The crop observations, economics, and yield data were indicated in the other extracted reports from this excrement which indicated that, the effect of the irrigation regime was not significant on flower number, fresh weight of flower, and dried stigma yield of saffron. In intercropping ratios, the highest flower number ($141.5 \pm 4.26$ m$^{-2}$), fresh weight of flower ($108.22 \pm 3.4$ g m$^{-2}$), and dried stigma yield ($4.1 \pm 0.01$ kg h$-1$) were observed in saffron mono-crop and approximately equaled by C3 treatment [61].

Based on these results, which were reported in previous article [61] and weed controlling by intercropping system in saffron and chickpea, it could be related to the nitrogen fixation by chickpea as a leguminous crops and saffron as economical crop which produce the suitable yield in this cropping system.

## Conclusion

According to the result, low irrigation regimes had a significant effect on weed density and biomass as well as on weed diversity. Low irrigation regimes reduced the density, biomass, and

diversity of weeds as compared to the conventional irrigation regimes through limiting the water availability to weeds. In addition, the intercropping system showed a significant effect on weed density and biomass and weed diversity in both irrigation regimes. Intercropping systems reduced weed density and biomass as compared to mono-crop. The weed diversity was reduced with saffron+chickpea intercropping bylimiting the recourse availability such as water, light, etc. consequently, low irrigation regimes and intercropping systems could be a proper strategy for weed control in the saffron field, especially in arid and semiarid regions. Future research in this area could investigate the potential of other leguminous crops for intercropping with saffron and explore the effects of different irrigation regimes on saffron yield and weed control. This could lead to the development of more sustainable and efficient saffron cultivation practices in water-limited regions.

## Author Contributions

**Supervision:** Majid Pouryousef.

**Writing – original draft:** Fatemeh Mohammadkhani.

**Writing – review & editing:** Fatemeh Mohammadkhani, Ali Reza Yousefi, Jose L. Gonzalez-Andujar.

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
