## [Decision Letter · Decision Letter 0]

17 Feb 2023

PONE-D-22-26195Weed community changes in saffron–chickpea intercropping under different irrigation managementPLOS ONE

Dear Dr. Pouryousef,

Thank you for submitting your manuscript to PLOS ONE. After careful consideration, we feel that it has merit but does not fully meet PLOS ONE’s publication criteria as it currently stands. Therefore, we invite you to submit a revised version of the manuscript that addresses the points raised during the review process.

ACADEMIC EDITOR: Manuscript English needs to improve.Provide some quantitative data in the abstract for a better understanding.Provide details such as the number of saffron plants in mono-cropping, nutrient management, and other agronomic practices followed in this experiment, what was the size of the main and subplot in this study.Provide the test names for normality and homogeneity of variance in the statistics section.Add crop observations, economics, and yield data in the result section to strengthen the manuscript.For more comments please see the manuscript.==============================

We look forward to receiving your revised manuscript.

Kind regards,

Prabhu Govindasamy, Ph.D.

Academic Editor

PLOS ONE

A clean copy of the edited manuscript (uploaded as the new *manuscript* file).

Additional Editor Comments:

• Manuscript English needs to improve.

• Provide some quantitative data in the abstract for a better understanding.

• Provide details such as the number of saffron plants in mono-cropping, nutrient management, and other agronomic practices followed in this experiment, what was the size of the main and subplot in this study.

• Provide the test names for normality and homogeneity of variance in the statistics section.

• Add crop observations, economics, and yield data in the result section to strengthen the manuscript.

• For more comments, please see the manuscript.

Reviewers' comments:

Reviewer's Responses to Questions

**Comments to the Author**

1. Is the manuscript technically sound, and do the data support the conclusions?

Reviewer #1: Yes

Reviewer #2: Yes

2. Has the statistical analysis been performed appropriately and rigorously? 

Reviewer #1: Yes

Reviewer #2: Yes

3. Have the authors made all data underlying the findings in their manuscript fully available?

Reviewer #1: Yes

Reviewer #2: Yes

4. Is the manuscript presented in an intelligible fashion and written in standard English?

Reviewer #1: Yes

Reviewer #2: Yes

5. Review Comments to the Author

Reviewer #1: Weed community changes in saffron–chickpea intercropping under different irrigation management-This research investigated irrigation regime and intercropping, effect on weed control and diversity. The research is unique nd would be of interest to PLOSE ONE readers.

Title is confusing looking like you used only intercrop of saffron and chickpea. But in this study you also used sole saffron and sole chickpea.

The English and scientific writing needs improvement.

There are several details missing to fully describe material & methods effectively: -how you calculate net plot in this study?

-nutrient management part is missing.

-What was the nutrient content in studied soil?

-Size of each main plot?

-You collected weed data at 75 DAS of chickpea. But by this time the weeds would have affected the saffron.

-How you checked normality and variance homogeneity of data?

-What was the objective to take sole chickpea here?

-Follow journal(PLOS ONE) pattern in all references.

-Italic all scientific names.

Specific comments:

-Full and short title both are same.

-Arrange key words in alphabetical order.

-Line17: Both in eight row. What does it mean?

-Line 36: Kg ha-1 make it in superscript.

-Line 37: Delete production.

-Line 50: This sentence need a complete revision to make it sound good.

-Line 72: Vicia faba in italic.

-Line 80-82: Remove too much result based references from introduction.

-Line 83: Add space in between "%" and "27".

-Line 85: Cicer arietinum in italic

-Line 93: Delete according to.

-Line 94: could be in place of be could.

-Line120: plant m-2 in place of plant m2.

-Line 125: How were the sample location selected in each plot?

-Line132 and 133: Eh and Ln are change in formula.

-Line239: Resource in place of recource.

-Line 254: et al??

The discussion part is poor and lacks proper explanation and reasoning.

To make their writing easier to read, the authors should improve it.

Reviewer #2: I have gone through the manuscript "Weed community changes in saffron–chickpea intercropping under different irrigation management". There study was carried out at semi-arid cold ecologies of Zanjan, Iran in the randomised block design. The manuscript explains an environment-friendly system of weed control. Considering the novelty, lack of information on saffron+chickpea intercropping system and the quality of MS, I feel that there are several strong reasons to accept the MS after moderate level of revisions. Addressing the below points will add more value to the MS:

1. Questions to be addressed -

· What is the effect of this system on saffron yield?

· What about the cost-effectiveness of this system (net profits and cost:benefit ratio)?

· How the crop management was done and what agronomic management was practiced. Corrections: Line 83 - %, Line 94 – could be, Line 239 – resource

2. Write saffron + chickpea instead of saffron - chickpea as "+" is more appropriate and internationally acceptable for intercropping systems.

3. The results in abstract needs to be written in quantitative terms. Merely writing the statements are not sufficient for a scientific paper.

4. Discussion part is poorly written. Please discuss the results in a more comprehensive manner.

6. PLOS authors have the option to publish the peer review history of their article (what does this mean?). If published, this will include your full peer review and any attached files.

Reviewer #1: No

Reviewer #2: No

---

## [Author Response · Author response to Decision Letter 0]

19 Mar 2023

Dear Dr. ....,

We would like to thank you for the letter, and the opportunity to submit a revised copy of this manuscript. We would also like to take this opportunity to express our thanks to the reviewers for the comments. We believe have resulted in an improved revised manuscript, which you will find uploaded alongside this document. The manuscript has been revised to address the reviewer's comments, detailed in your letter properly as explained below.

---

## [Decision Letter · Decision Letter 1]

2 May 2023

PONE-D-22-26195R1Weed community changes in saffron–chickpea intercropping under different irrigation managementPLOS ONE

Dear Dr. Pouryousef,

Thank you for submitting your manuscript to PLOS ONE. After careful consideration, we feel that it has merit but does not fully meet PLOS ONE’s publication criteria as it currently stands. Therefore, we invite you to submit a revised version of the manuscript that addresses the points raised during the review process.

We look forward to receiving your revised manuscript.

Kind regards,

Prabhu Govindasamy, Ph.D.

Academic Editor

PLOS ONE

Journal Requirements:

Additional Editor Comments:

Manuscript English is not up to the acceptable level. In abstract and conclusion need to do more work, see my attached file. Provide yield data.

Reviewers' comments:

Reviewer's Responses to Questions

**Comments to the Author**

1. If the authors have adequately addressed your comments raised in a previous round of review and you feel that this manuscript is now acceptable for publication, you may indicate that here to bypass the “Comments to the Author” section, enter your conflict of interest statement in the “Confidential to Editor” section, and submit your "Accept" recommendation.

Reviewer #1: All comments have been addressed

Reviewer #2: All comments have been addressed

2. Is the manuscript technically sound, and do the data support the conclusions?

Reviewer #1: Yes

Reviewer #2: Yes

3. Has the statistical analysis been performed appropriately and rigorously? 

Reviewer #1: Yes

Reviewer #2: Yes

4. Have the authors made all data underlying the findings in their manuscript fully available?

Reviewer #1: Yes

Reviewer #2: Yes

5. Is the manuscript presented in an intelligible fashion and written in standard English?

Reviewer #1: Yes

Reviewer #2: Yes

6. Review Comments to the Author

Reviewer #1: Study covered weed community changes in saffron–chickpea intercropping under different irrigation management-The research is unique and would be of interest to PLOSE ONE readers. However, there are minor corrections which should be addressed for improving quality of manuscript.

Full and short title are same.

Nutrient management part is still missing in material & method section.

Line 85: 27% and 15%

Line 89: correct sentence.

Line 89: write scientific name in italic.

Line 93: Remove according to.

Line 95: Saffron

Line 99: saffron+chickpea

Line 114: Correct alignment.

Line 116: m2 in superscript.

Line 121 : replace data with date.

Line 125: Give the stages when you applied irrigations.

Line: 129: figure.

Line 171: use multiplication symbol instead of *.

Line 191: Fig 3a & 3b.

Discussion part need to improve.

Yield data is missing.

Reviewer #2: The authors have done sufficient revisions. However, the minor English language related issues and formatting of references as per the journal format may be done. The manuscript may be accepted.

7. PLOS authors have the option to publish the peer review history of their article (what does this mean?). If published, this will include your full peer review and any attached files.

Reviewer #1: No

Reviewer #2: No

While revising your submission, please upload your figure files to the Preflight Analysis and Conversion Engine (PACE) digital diagnostic tool, https://pacev2.apexcovantage.com/. PACE helps ensure that figures meet PLOS requirements. To use PACE, you must first register as a user. Registration is free. Then, login and navigate to the UPLOAD tab, where you will find detailed instructions on how to use the tool. If you encounter any issues or have any questions when using PACE, please email PLOS at figures@plos.org. Please note that Supporting Information files do not need this step.<quillbot-extension-portal></quillbot-extension-portal><quillbot-extension-portal></quillbot-extension-portal><quillbot-extension-portal></quillbot-extension-portal><quillbot-extension-portal></quillbot-extension-portal>

---

## [Author Response · Author response to Decision Letter 1]

7 May 2023

Dear Dr. ....,

We would like to thank you for the letter, and the opportunity to submit a revised copy of this manuscript. We would also like to take this opportunity to express our thanks to the reviewers for the comments. We believe have resulted in an improved revised manuscript, which you will find uploaded alongside this document. The manuscript has been revised to address the reviewer's comments, detailed in your letter properly as explained below.

---

## [Decision Letter · Decision Letter 2]

17 May 2023

Weed community changes in saffron–chickpea intercropping under different irrigation management

PONE-D-22-26195R2

Dear Dr. Majid Pouryousef,

We’re pleased to inform you that your manuscript has been judged scientifically suitable for publication and will be formally accepted for publication once it meets all outstanding technical requirements.

Kind regards,

Prabhu Govindasamy, Ph.D.

Academic Editor

PLOS ONE

Additional Editor Comments (optional):

Reviewers' comments:

Reviewer's Responses to Questions

**Comments to the Author**

Reviewer #1: All comments have been addressed

2. Is the manuscript technically sound, and do the data support the conclusions?

Reviewer #1: Yes

3. Has the statistical analysis been performed appropriately and rigorously? 

Reviewer #1: Yes

4. Have the authors made all data underlying the findings in their manuscript fully available?

Reviewer #1: Yes

5. Is the manuscript presented in an intelligible fashion and written in standard English?

Reviewer #1: Yes

6. Review Comments to the Author

Reviewer #1: (No Response)

7. PLOS authors have the option to publish the peer review history of their article (what does this mean?). If published, this will include your full peer review and any attached files.

Reviewer #1: No

---

## [Editor Report · Acceptance letter]

19 May 2023

PONE-D-22-26195R2 

Weed community changes in saffron+chickpea intercropping under different irrigation management 

Dear Dr. Pouryousef:

I'm pleased to inform you that your manuscript has been deemed suitable for publication in PLOS ONE. Congratulations! Your manuscript is now with our production department. 

Kind regards, 

on behalf of

Dr. Prabhu Govindasamy 

Academic Editor

PLOS ONE